# An optimised faecal microRNA sequencing pipeline reveals fibrosis in *Trichuris muris* infection

Emma Layton[1,2,3], Sian Goldsworthy [4], EnJun Yang[3], Wei Yee Ong[3,5], Tara E. Sutherland [6], Allison J. Bancroft[1,2,7], Seona Thompson [1,2,7], Veonice Bijin Au [3], Sam Griffiths-Jones [4], Richard K. Grencis [1,2,7] ✉, Anna-Marie Fairhurst [3,8] ✉ & Ian S. Roberts [1,2] ✉

The intestine is a site of diverse functions including digestion, nutrient absorption, immune surveillance, and microbial symbiosis. Intestinal micro-RNAs (miRNAs) are detectable in faeces and regulate barrier integrity, host-microbe interactions and the immune response, potentially offering valuable non-invasive tools to study intestinal health. However, current experimental methods are suboptimal and heterogeneity in study design limits the utility of faecal miRNA data. Here, we develop an optimised protocol for faecal miRNA detection and report a reproducible murine faecal miRNA profile in healthy mice. We use this pipeline to study faecal miRNAs during infection with the gastrointestinal helminth, *Trichuris muris*, revealing roles for miRNAs in fibrosis and wound healing. Intestinal fibrosis was confirmed in vivo using Hyperion® imaging mass cytometry, demonstrating the efficacy of this approach. Further applications of this optimised pipeline to study host-microbe interactions and intestinal disease will enable the generation of hypotheses and therapeutic strategies in diverse contexts.

MicroRNAs (miRNAs) function as post-transcriptional regulators of gene expression in the cell cytoplasm of nematodes, plants and animals[1–3]. Mature miRNAs can also be detected extracellularly in bodily fluids including serum, saliva, breast milk, cerebrospinal fluid, seminal fluid, urine, and faeces[4,5]. Here, encapsulation within the phospholipid membranes of extracellular vesicles (EVs) enhances their extracellular stability[6]. These EVs are decorated with proteins and lipids that facilitate their function as communication molecules and enable the transport of diverse cargo including miRNAs between different tissues[7].

In the mammalian intestine, miRNAs perform essential roles in barrier integrity, metabolism, regulation of the mucosal immune response and in microbial homoeostasis[5,8–13]. The miRNAs produced by intestinal epithelial cells are detectable in EVs in faeces, and offer a promising non-invasive tool to study intestinal disease[14–17]. Previous work has employed microarrays and qPCRs to study faecal miRNAs, primarily

[1]Division of Infection, Immunity and Respiratory Medicine, Faculty of Biology, Medicine and Health, University of Manchester, Manchester, UK. [2]The Lydia Becker Institute of Immunology and Inflammation, Faculty of Biology, Medicine and Health, University of Manchester, Manchester, UK. [3]Institute of Molecular and Cell Biology, Agency for Science, Technology and Research, Singapore, Singapore. [4]Division of Evolution, Infection and Genomics, Faculty of Biology, Medicine and Health, University of Manchester, Manchester, UK. [5]Microbiology and Immunology Department, Yong Loo Lin School of Medicine, National University of Singapore, Singapore, Singapore. [6]Institute of Medical Sciences, School of Medicine, Medical Sciences and Dentistry, University of Aberdeen, Aberdeen, UK. [7]The Wellcome Trust Centre for Cell-Matrix Research, University of Manchester, Manchester, UK. [8]Singapore Immunology Network (SIgN), Agency for Science, Technology and Research, Singapore, Singapore. ✉e-mail: richard.grencis@manchester.ac.uk; annamarie@imcb.a-star.edu.sg; i.s.roberts@manchester.ac.uk

in the contexts of colorectal cancer and inflammatory bowel disease (IBD)[18–20]. In gastrointestinal cancers and IBD, alterations in specific miRNAs have been identified as novel diagnostic tools and indicators of disease severity and prognosis[18,21,22]. However, substantial heterogeneity in study design, including variations in sample storage, miRNA isolation, and data analysis, limits the reproducibility and utility of these findings[18]. More recently, faecal miRNA profiles in humans and mice were examined using high-throughput small RNA sequencing[14–17,23,24]. However, there remains a limited understanding of the reproducibility of miRNA profiles in faecal samples across studies, or if faecal miRNA profiles are stable in health and homoeostasis.

We addressed these limitations by developing an optimised faecal small RNA sequencing pipeline and performing a meta-analysis of recently published small RNA sequencing data from murine faecal samples. As a result, we establish best practices for this technique and identify a core faecal miRNA profile that is consistent across laboratory practices. Furthermore, our current study reports the first longitudinal analysis of murine faecal miRNAs in healthy mice. We then demonstrate the utility of our pipeline in hypothesis generation by analysing changes in faecal miRNAs due to chronic *Trichuris muris* (mouse whipworm) infection.

*T. muris* is a model of the human gastrointestinal whipworm, *Trichuris trichiura*, which is estimated to infect over 500 million individuals globally and causes chronic diarrheal disease predominantly in children[25,26]. During pathogenesis, *T. muris* disrupts the intestinal epithelium, perturbs the microbiota, and induces a Th1 or Th2-polarised mucosal immune response, each of which are processes that are regulated by miRNAs in the intestine[5,8–11,27,28]. In *T. muris* infection, the generation of a Th1-type immune response is associated with chronicity and the inability to clear the infection[28]. Conversely, mice that are resistant to infection generate a Th2-polarised response that facilitates wound healing, tissue remodelling and expulsion[28]. Here, we demonstrate that chronic infection induces the differential expression of miRNAs associated with wound healing, barrier integrity, and fibrosis. These findings suggest that miRNAs may be a route by which tissue damage is repaired in a Th1-polarised chronic infection, despite the absence of the typical tissue remodelling anti-helminth Th2 cytokines.

## Results

### An optimised library preparation pipeline improves miRNA yields

To date, there have been a limited number of studies examining faecal miRNA profiles by small RNA sequencing. Furthermore, small RNA library-building kits are not designed for this atypical tissue type. Therefore, we initially optimised the faecal small RNA sequencing protocol by testing different storage conditions for faecal pellets. Notably, we found that storing faecal pellets at 4 °C in RNAlater stabilising buffer was sufficient to yield high-quality total RNA (as evidenced by electropherogram analysis) obviating the need for snap freezing (Supplementary Fig. 1). Due to the high proportion of small RNAs in faecal samples relative to other tissues, we also opted for a lower number of PCR cycles during library building to facilitate the clear separation and visualisation of bands on the size selection gel (see "Methods" and Supplementary Fig. 1).

We next assessed whether our optimised protocols resulted in an improved miRNA sequence yield by conducting a meta-analysis of our data together with the raw data extracted from recently published studies in mice[14–16]. The inclusion criteria for studies in the meta-analysis were murine faecal small RNA sequencing data from C57BL/6 mice that were publicly available and published during the study period (up until October 2024). The raw data from these studies was re-analysed with the bioinformatics pipeline described here (see "Methods") to enable the comparison of sequencing outputs.

Our optimised methods yielded the highest mean normalised miRNA reads per sample, surpassing the study with the next highest yield, He et al.[15], by 23.5% (Table 1). Total RNA is typically recommended as input for small RNA library construction and small RNA is later size-selected for, however Tomkovich et al.[14], extracted only small RNAs from faecal samples for library preparation. This group also utilised the lower throughput MiSeq sequencing machine, and subsequently obtained the lowest number of average raw sequencing reads in our analysis. Across all studies, a substantial proportion of small RNA sequencing reads were discarded due to undesired read length (less than 18, or greater than 26 nucleotides). The proportion of tRNA reads remained consistently low across datasets (<1.23%), whereas rRNA comprised a more substantial proportion, ranging from 8.4% to 57.95% of filtered reads. Among the datasets analysed, the study by He et al.[15], detected the highest number of miRNA species (121) closely followed by our current study (120). These findings suggest that our sample and library preparation conditions are optimal, yielding both a high number of miRNA reads and a diverse repertoire of detected miRNA species.

### Faecal miRNA profiles of healthy mice are reproducible across studies

Our optimised methods demonstrate effective miRNA isolation from faecal samples. Building on these observations, we assessed the reproducibility of faecal miRNA profiles in healthy wild-type mice across the different studies. This was evaluated by comparing the most abundant miRNAs from the baseline healthy control group in the current study (I0), to the non-immunised group in Liu et al.[16], and the healthy untreated DSS0 group in He et al.[15]. Tomkovich et al.[14], did not include a wild-type mouse group and was therefore excluded. Initially, the top 20 miRNAs in each study measured by mean normalised counts were selected. The median ranking of these miRNAs across the three studies was then calculated to visualise the most abundant and consistently detected miRNAs (Fig. 1). Thirty unique miRNAs were found in the top 20 of the three studies, and 12 of the top 20 were consistent across all three studies. The most abundant miRNA in all studies was miR-21a-5p. Additionally, the miR-200 family (including miR-200a, 200b, and 200c) and members of the highly conserved let-7 family (comprising let-7a, let-7b, let-7c, let-7f, let-7g, and let-7i) were reproducibly detected at a high abundance.

### miRNA profiles of healthy mice are reproducible over time

The dynamics of faecal miRNA expression over time have not been previously reported. Our study profiles faecal miRNAs in healthy wild-type mice at three time points (I0, N21 and N35) (Fig. 2a). The top 20 most abundant miRNAs were remarkably consistent across these time points with the top three most abundant miRNAs (miR-21a-5p, miR-192-5p and miR-148a-3p) remaining constant (Fig. 2b). Furthermore, 18 out of the top 20 most abundant miRNAs were reproducibly detected at each time point. While the miRNA composition is highly conserved, the expression of the most abundant miRNAs does change significantly over time (Fig. 2c–f). Interestingly, we observed that miRNAs in the same family show the same expression pattern over time. For example, all members of the let-7 family increased in abundance over time (Fig. 2c) and all members of the miR-200 family decreased (Fig. 2e). To ensure that this trend was not due to the sharing of non-unique reads across family members, this analysis was reproduced by counting only unique reads and the observed trend remained (Supplementary Fig. 2). Some of the reproducibly abundant miRNAs show non-linear changes in expression and decrease in abundance after 21 days before increasing again after 35 days, or vice versa (Fig. 2d, f).

### Faecal miRNAs are differentially expressed by chronic infection

Our work demonstrates small RNA sequencing of faecal miRNAs to be effective and robust to laboratory practices and environmental

**Table 1 | Library building techniques and mapping statistics across mouse faecal small RNA sequencing studies**

| Sample information | Liu et al. (n = 15) | Tomkovich et al. (n = 21) | He et al. (n = 11) | Layton et al. (current n = 31) |
|---|---|---|---|---|
| Faecal storage prior to extraction | Snap frozen | Snap frozen | N.S. | 4 °C in RNAlater immediate extraction |
| Extraction kit and RNA enrichment | MirVana total RNA | MirVana small RNA | MirVana total RNA | MirVana total RNA |
| Library kit | Nextflex | NEBnext | NEBnext | NEBnext |
| Sequencing machine | NextSeq | MiSeq | HiSeq | HiSeq |
| Raw reads (range) | 33,775,633 (20,170,514–55,664,440) | 794,663 (354,995–1,595,598) | 21,620,070 (19,922,251–23,545,474) | 26,246,329 (13,603,079–39,286,425) |
| Reads passing QC & filtering | 25.14% (8,530,128) | 29.24% (233,169) | 41.54% (8,910,852) | 38.8% (10,267,557) |
| tRNA reads after filtering | 0.21% (17,868) | 1.23% (3,156) | 0.62% (55,163) | 0.27% (27,905) |
| rRNA reads after filtering and tRNA removal | 8.4% (729,226) | 57.95% (132,593) | 18.91% (1,775,245) | 24.56% (2,581,720) |
| Reads mapping to mouse genome after filtering, tRNA and rRNA removal | 16.75% (1,310,227) | 45.11% (41431) | 21.17% (1,367,993) | 25.48% (1,922,223) |
| Raw miRNA reads as % of total reads passing QC | 0.19% | 0.62% | 0.40% | 0.46% |
| Raw miRNA reads (range) | 16,562 (1444–51,429) | 1,351 (174–4113) | 30,262 (16,812–47,642) | 43,989 (10,172–122,794) |
| Normalised miRNA reads (range) | 10,735 (9434–14,099) | 878 (699–1083) | 28,522 (21,285–37,969) | 35,225 (28,948–42,443) |
| miRNA species detected at >10 raw counts in 5 samples | 83 | 25 | 121 | 120 |

Mapping statistics displayed are the average across all samples in each study after re-analysis of the published raw data with the bioinformatics pipeline developed in this study.
N.S. not stated.

variables. To evaluate whether this technique can generate testable hypotheses in disease contexts, we evaluated changes in faecal miR-NAs as a consequence of intestinal helminth infection. *T. muris* is a murine gastrointestinal helminth that invades the intestinal epithelium, inducing damage and a mucosal immune response[28]. Here, miRNAs have roles in barrier integrity and in the polarisation of the T helper type immune response[8,29]. Given that *T. muris* disrupts the intestinal epithelium and the polarisation of the T helper response is critical in infection outcome, we investigated whether faecal miRNAs were differentially expressed by chronic *T. muris* infection[30,31].

Faecal miRNA profiles of infected and naïve mice were compared on day 21 and day 35 of the experiment, when the worms are L3/L4 larvae and adults, respectively[28]. At day 21 one miRNA was differentially expressed, miR-7a-5p was significantly upregulated (adj $p = 0.029$). The most significantly differentially expressed miRNA by day 35 of infection was miR-6239 (adj $p = 0.001$), which is a recently discovered miRNA. However, on further examination, we observed that the small RNA reads that map to this miRNA locus map with a single nucleotide mismatch, and these reads also map to mouse 5S rRNA with a single nucleotide mismatch at a different position (Supplementary Fig. 3). Since we cannot be confident that this is a genuine miRNA, it was excluded from further analysis. There were seven remaining high-confidence differentially expressed miRNAs at day 35 by chronic *T. muris* infection: three were significantly upregulated (miR-200c, miR-23b and miR-5119) and four were significantly downregulated (miR-26b, miR-29a, miR-103 and miR-6966). Principal component analysis of all samples revealed that, despite some variability in overall faecal miRNA profiles between the individuals within each group, the majority of the variance is explained by infection status by day 35 (see Supplementary Fig. 4).

To gain a functional insight into the role of the differentially expressed miRNAs at day 35 post-infection, pathway analysis was performed on their predicted mRNA targets (Fig. 3a). This analysis highlighted a potential role for these miRNAs in the development of fibrosis, with other enriched pathways including wound healing, epithelial adherens junction signalling and tight junction signalling. These results suggest that host miRNAs could be regulating the damage response caused by the invasion of *T. muris* into the epithelium.

**Differentially expressed miRNAs increase collagen in fibroblasts**
Fibroblasts are key effector cells in fibrosis that undergo conversion to pro-fibrotic myofibroblasts when stimulated by TGF-β[32]. To further understand the functional consequences of the *T. muris*-induced miRNA changes in fibrosis, the top three most differentially expressed miRNAs at day 35 (miR-26b, miR-29a and miR-200c) were transfected into TGF-β stimulated 3T3 fibroblasts in vitro. Changes in the expression of the myofibroblast markers alpha-smooth muscle actin (α-SMA, encoded by *Acta2* gene) and collagen I (encoded by *Col1* gene) were assessed at both the transcript and protein levels[33]. A Cy3-labelled negative control miRNA (miR-ctrl) was used to account for any possible effects of the transfection process itself.

The transfection of miR-29a or miR-200c resulted in striking changes in *Col1* expression (Fig. 3b). The miR-29a mimic inhibited *Col1* expression in TGF-β treated 3T3 fibroblasts, and conversely the miR-29a inhibitor led to a 13-fold increase in *Col1* expression. An opposing phenotype was observed for miR-200c, where the addition of the mimic induced a significant upregulation of *Col1* expression (Fig. 3b). The changes in *Col1* expression were also confirmed at the protein level with confocal microscopy (Fig. 3c). The miRNA mimics and inhibitors for these three miRNAs did not significantly influence *Acta2* expression compared to the miR-ctrl transfected cells (Supplementary Fig. 5). This suggests that the miRNA-induced changes in collagen production within fibroblasts may be independent of myofibroblast formation in this context.

In parallel to the experiments described above, cytokines present in the cell culture supernatant from the transfected fibroblasts

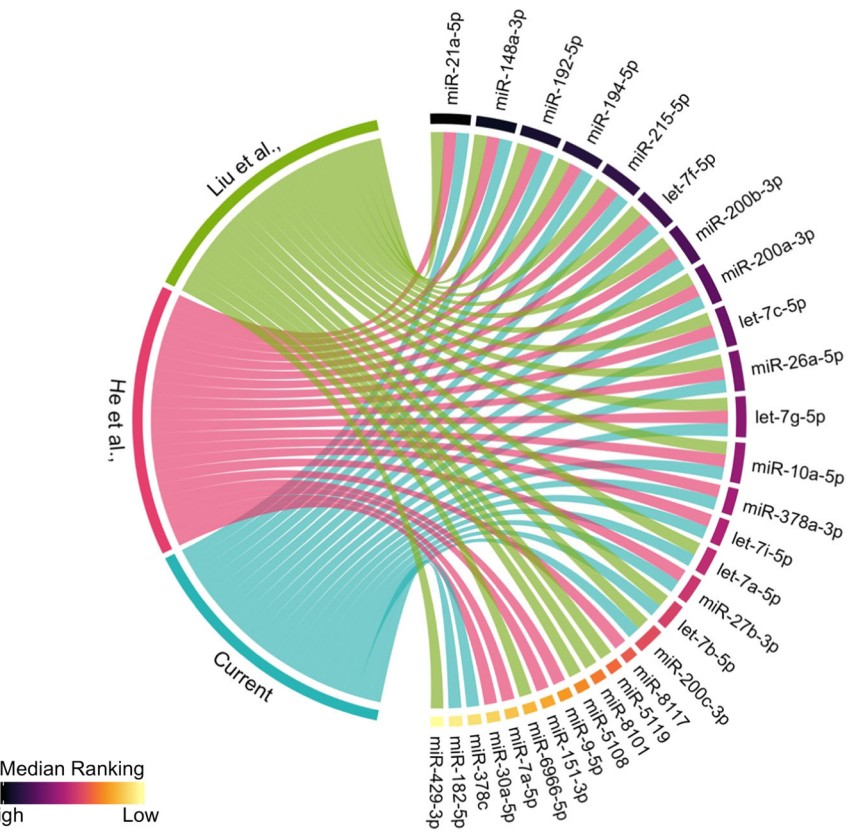

**Fig. 1 | Faecal miRNA profiles of healthy wild-type mice are reproducible across studies.** The top 20 miRNAs measured by mean normalised counts for each study are represented as 20 chords, and each study represented in a different colour. On the right-hand side, miRNAs are sorted clockwise by their median ranking across the three studies. Liu et al. represents the non-vaccinated group of mice[16] ($n = 5$, green) and He et al. represents the samples taken at day 0 prior to dextran sulphate sodium treatment (DSS0)[15] ($n = 5$, pink). Current represents the samples taken from the infected group prior to infection at the baseline time point on day 0 (I0) in the current study ($n = 7$, blue).

were profiled by Luminex analysis. These analyses revealed that the addition of the miR-200c mimic significantly decreased the production of the Th2-priming cytokine, C-C motif chemokine ligand 2 (CCL2) by over 40%, while inducing the production of the Treg polarising cytokine, leukaemic inhibitory factor (LIF) (Fig. 3d)[34–36]. Taken together we show that decreased miR-29a and increased miR-200c expression in chronic infection are associated with increased collagen production, a key feature of fibrosis. In addition, we predict that increased miR-200c could also alter the immune response through the regulation of T cell polarisation via CCL2 and LIF.

### Evidence of fibrosis in the caecum in *T. muris* infection

After demonstrating the differential expression of miRNAs involved in fibrosis and collagen production, we went on to examine the presence of fibrosis in vivo following *T. muris* infection. We used Hyperion® imaging mass cytometry to characterise pathology, which combines cutting-edge time-of-flight cytometry technology with immunohistochemical staining. Sections of caecum tissue obtained from healthy C57BL/6 and *T. muris*-infected mice were assessed with a panel of fibrosis markers (Fig. 4a). Extensive remodelling was observed in infected tissue, with a pronounced increase in the expression of hyaluronan (yellow), a glycosaminoglycan regulated by TGF-$\beta$ and associated with fibrosis in several diseases including IBD[37,38]. In contrast, hyaluronan expression was faint in naïve tissue, and the crypts were predominantly covered with collagen VI (blue). Heparan sulphate (magenta) is a key structural component of the extracellular matrix that modulates signalling by fibrosis regulators, such as TGF-$\beta$ and bone morphogenic proteins[39,40]. A discernible layer of heparan

sulphate was evident at the base of the crypts in tissue from chronic infection, in contrast to the naïve tissue (Fig. 4a and Supplementary Fig. 6).

Most strikingly, infected mice displayed a substantially thicker layer of collagen I (green) in the submucosa compared to their uninfected counterparts. In agreement with our previous findings that miR-29a and miR-200c increased collagen I but did not significantly influence *Acta2* expression in fibroblasts, the region of increased collagen I deposition does not co-stain for α-SMA (Fig. 4b–d). These data together support our predicted involvement of faecal miRNAs in collagen production and the development of fibrosis in the caeca of mice with a chronic *T. muris* infection.

## Discussion

Small RNA sequencing of faecal miRNAs provides a non-invasive method to explore the transcriptional state of the intestine in health and disease. Compared to conventional methods such as colonoscopies and endoscopies, it is less invasive, does not require specialised administration, and is more accessible for research, providing a practical tool for monitoring intestinal health. However, faecal miRNA sequencing protocols and the typical miRNA content of faecal samples remain poorly described. Here, we provide a comprehensive description of faecal miRNA sequencing from sample collection through to analysis and demonstrate a stable faecal miRNA profile in healthy mice. We then employ this tool to describe a role for host miRNAs in the host-pathogen relationship during infection with the intestinal helminth, *T. muris*, a model of the human intestinal helminth, *T. trichiura*[28].

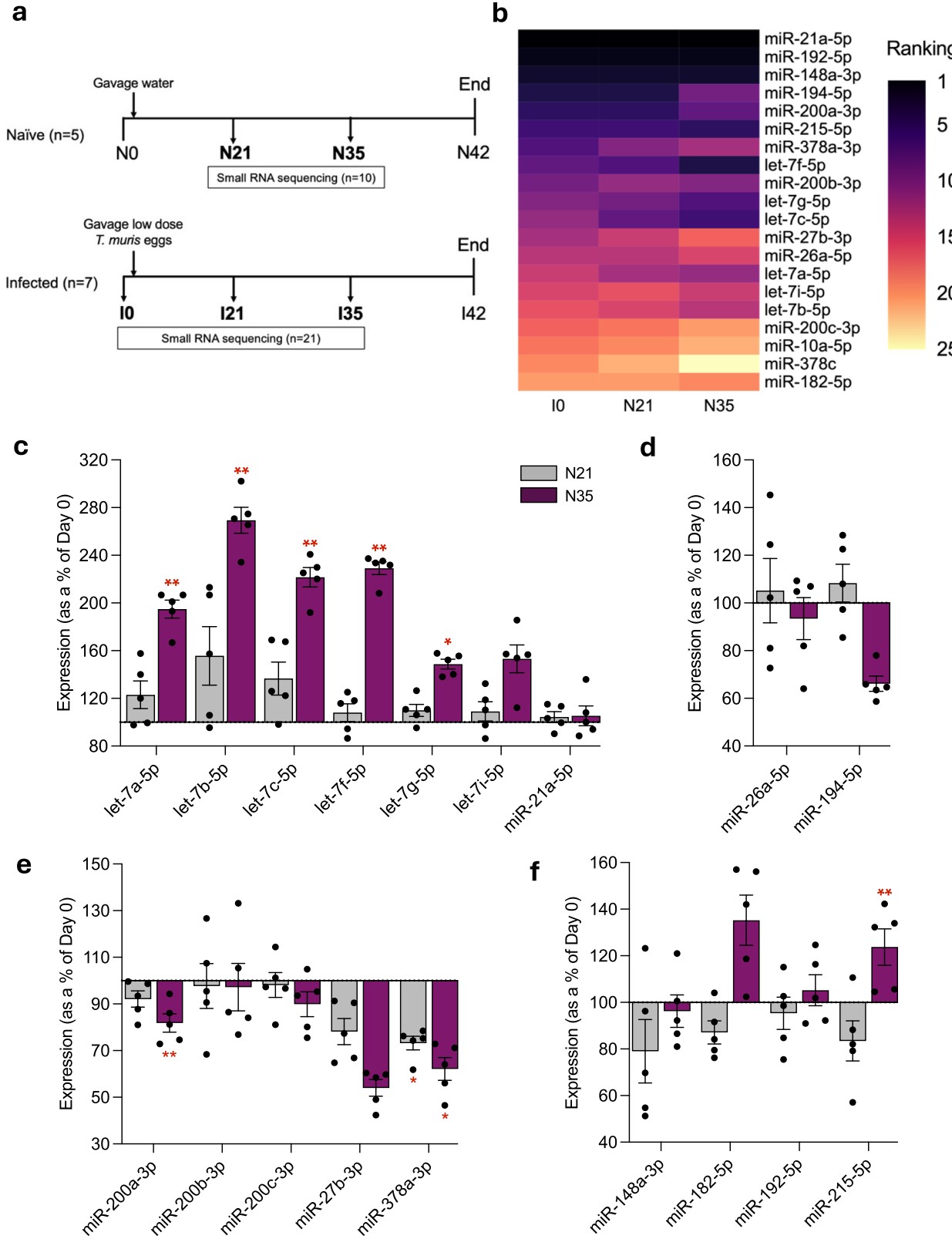

Small RNA sequencing of faecal samples to study miRNAs has unique challenges due to the high proportion of degraded RNA species such as tRNA and rRNA. However, miRNAs themselves are stable in faeces and can be protected from degradation by encapsulation in EVs[5]. Despite inter-study variables such as housing conditions, diet, exposure to microbes, sample processing and extraction methods, we identify a remarkably reproducible core faecal miRNA profile in healthy mice across several studies (consisting of miR-21a, miR-148a, miR-192, miR-194, miR-215 and members of the let-7 and miR-200 families). In this study, we found that in agreement with previous literature, the storage of faecal pellets at 4 °C and the use of stabilising buffers is sufficient to prevent miRNA degradation in faecal samples[41,42]. In doing so, we were able to increase the miRNA read yield compared to previous studies. We also identify key areas

**Fig. 2 | Dynamics of most abundant faecal miRNAs in healthy mice over time.**
**a** Representation of sample collection workflow and nomenclature of faecal samples for small RNA sequencing from naïve mice and mice infected with a low dose of *T. muris*. **b** Heatmap comparing the ranking of the top 20 miRNAs measured by mean normalised counts in the healthy control group at the baseline (I0) to the other naïve groups control groups at day 21 and 35 (N21 and N35). **c–f** Expression of miRNAs on day 21 and day 35 (N21 and N35) shown as a percentage change from the value at day 0 (I0). miRNAs are clustered into sub-plots based on expression pattern over time. Mean and SEM are plotted. Statistical significance was calculated on the normalised counts for each group by two-way ANOVA with Geisser-greenhouse correction and Tukey's multiple comparisons test. Significance values displayed are for each time point versus I0. **c** miRNAs that increase incrementally, **d** miRNAs that increase at day 21 before decreasing at day 35, **e** miRNAs that decrease incrementally and **f** miRNAs that decrease at day 21 before increasing again at day 35. I0 = 7, N21 = 5 and N35 = 5 individuals. * = adj. $p < 0.05$, ** = adj. $p < 0.01$. Exact P values can be found in the Source Data file or Supplementary Data 2 (for **b**).

for further optimisations to this technique, such as a more effective exclusion of rRNA fragments prior to library preparation, and the improvement of the size selection methods implemented during library preparation. Such optimisations will further improve the sensitivity and cost-effectiveness of this method and facilitate its wider implementation as a hypothesis-generating tool.

We hypothesised that intestinal miRNAs would have a role in regulating the transcriptional response to *T. muris* infection. To establish infection, *T. muris* must burrow into and attach to the intestinal epithelium. In doing so, this causes damage that must be restrained to maintain the health of the host[43–45]. However, the orchestration of tissue repair in chronic infections characterised by low expression of classical wound healing Th2 cytokines, remains elusive. We identified several differentially expressed miRNAs during chronic infection, including established fibrosis regulators such as miR-29a and miR-200c[46–49]. We validate that the changes in these miRNAs during infection enhance collagen I production in fibroblasts, key mediators of fibrosis. Mechanistically, miR-200c likely exerts its regulatory effect by targeting GATA binding protein 4 (GATA4), a known repressor of collagen I in fibroblasts[50,51]. Conversely, miR-29a targets collagen I and impacts its expression directly[52,53]. Altered host miRNA expression may therefore regulate the wound healing response and induce fibrosis independently of Th2 cytokines. Furthermore, our findings suggest a mechanism for helminth-induced intestinal pathology via the perturbation of host miRNAs.

In addition to its impact on collagen I production, miR-200c regulated fibroblast cytokine secretion. In TGF-β-treated fibroblasts, miR-200c significantly decreased the secretion of CCL2, and increased the secretion of LIF. CCL2 is a potent Th2-type cytokine that is produced by dendritic cells upon stimulation of the TLR-MyD88-NF-kB pathway by *T. muris* excretory/secretory products[36,54]. This pathway also exists in fibroblasts, and MyD88 is a validated target of miR-200c, which suggests a potential mechanism for the observed inhibition of CCL2 by miR-200c (Supplementary Fig. 7)[55].

LIF is a member of the IL-6 cytokine family, and in the presence of TGF-β, high LIF levels and low IL-6 levels promote the differentiation of naïve T cells into Tregs; whilst low LIF levels and high IL-6 favours Th17 cell differentiation[34,35]. The induction of LIF by miR-200c may be attributed to miR-200c targeting of a LIF repressor, zinc finger E-box-binding homeobox 1 (ZEB1)[56,57]. Additionally, miR-200c directly targets IL-6 and its signalling pathways and reduces CCL2, an inducer of IL-6[58,59]. Taken together this data suggests that miR-200c shifts the Treg/Th17 cell balance towards anti-inflammatory Treg production and potentially limits immunopathology during chronic infection (Supplementary Fig. 8).

In addition to its roles in collagen production and cytokine secretion, miR-29a has several additional functions that may be relevant in *T. muris* infection. For example, miR-29a represses Th1-polarising transcription factors T-bet and Eomes, and negatively regulates IFN-γ production in T cells[60,61]. Furthermore, miR-29a downregulates the tight junction protein, claudin-1 in the intestinal epithelium and may facilitate stem cell-mediated regeneration of intestinal tissue following injury[62,63]. Therefore, reduced expression of miR-29a in *T. muris* infection may drive Th1-type immune responses and limit the host's anti-helminth proliferative response that can expel the parasite from the epithelium[64].

Hyperion mass cytometry imaging revealed significant fibrotic remodelling in the caecum as a consequence of infection, with changes in collagen I, hyaluronan and heparin sulphate expression. Intestinal fibrosis is not well described as a consequence of helminth infection, however, liver fibrosis is a common complication of infection with *Schistosomes*[65,66]. Notably, *Schistosoma japonicum*-induced liver fibrosis is also characterised by reduced miR-29a expression that correlates with fibrosis severity in humans, suggesting a potential common mechanism by which fibrosis is induced during infection with distinct helminth species[65]. Whilst our data suggests roles for host miRNAs in the development of pathology during *T. muris* infection, additional experiments, such as miRNA overexpression or knockdown are needed to confirm this in vivo. Exploring the roles of these miRNAs in other cell types, such as intestinal epithelial cells, would also provide a more comprehensive understanding of their function in *T. muris* pathology.

The mechanisms through which infection alters faecal miRNAs remain unclear. However, miRNAs are conserved across helminths and mammals, and helminths produce EVs that contain miRNAs and the Argonaute endonuclease effector protein[67–69]. Furthermore, mouse-derived miRNAs are abundant in *T. muris* EVs, indicating a potential to alter the host miRNA composition and transcriptional profile in pathogenesis[69]. However, additional research is required to determine the molecular mechanisms of such inter-species interactions. Endogenous miRNA expression, including that of miR-29a and miR-200c, can be influenced by dietary fat content[23,70–73]. Notably, dietary fat content can also influence the outcome of *T. muris* infection[74]. Dietary plants also contain exogenous plant miRNAs that are encapsulated in EVs and can modulate both the microbiota and the host, shaping the inflammatory response[12,75]. Together, these findings suggest promising therapeutic avenues to potentially modulate intestinal miRNAs through dietary interventions.

In conclusion, our study demonstrates the robustness of faecal miRNA profiling through small RNA sequencing as a non-invasive methodology for studying intestinal disease. By employing this tool to study *T. muris* infection, we identify potential roles for miRNAs, such as miR-29a and miR-200c, in shaping the immune response and contributing to intestinal fibrosis during chronic infection. Helminths, despite being recognised as pathogens that cause significant morbidity, paradoxically offer therapeutic potential in the treatment of inflammatory diseases due to their immunomodulatory properties[76,77]. Furthering understanding of the molecular underpinnings of the host-pathogen interaction may facilitate the refinement of helminth-inspired therapies for inflammatory diseases, possibly by disentangling the desirable anti-inflammatory effects from the potentially harmful pro-fibrotic effects of infection. Furthermore, fibrosis is a key contributor to tissue dysfunction and disease progression in multiple chronic inflammatory conditions of the intestine, and so the potential of specific faecal miRNAs to act as non-invasive biomarkers will be of wide interest[78]. Given the critical role of intestinal health in overall well-being, we anticipate that broader applications of this faecal small RNA sequencing pipeline will enable hypothesis generation and discovery in diverse contexts, including in infections, inflammatory conditions, microbial dysbiosis, and cancer.

## Methods
### Ethics statement
Experiments involving mice were performed in accordance with the United Kingdom's Animals (Scientific Procedures) Act 1986 under the

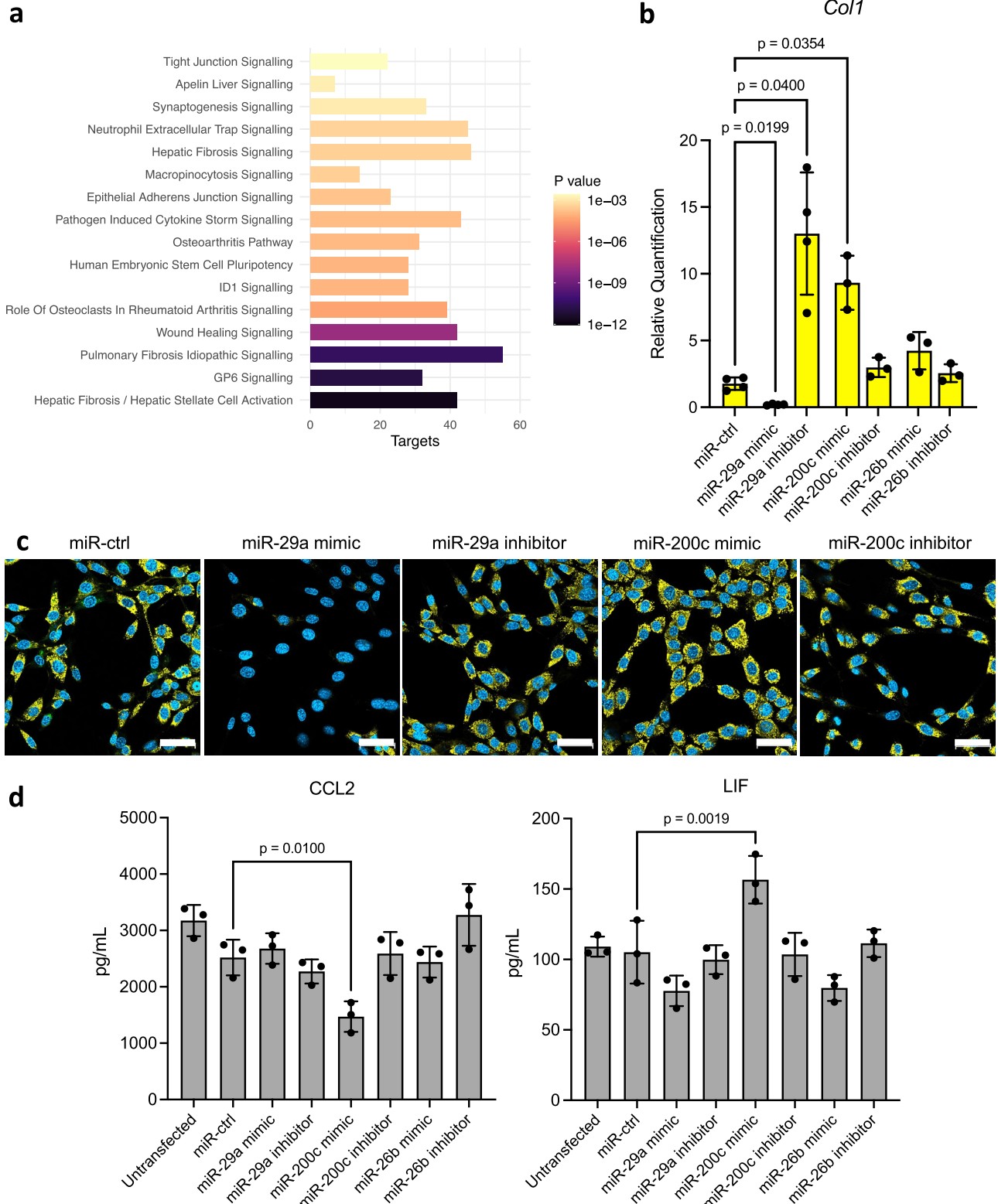

Home Office project license PO43A3082 and after local Animal Welfare and Ethical Review Body approval.

**Animals and sampling**

Male C57BL/6 mice of 6–8 weeks of age (Charles Rivers laboratories, UK) were co-housed for 2 weeks prior to the beginning of the experiment. Mice were infected by oral gavage with a low dose (-25 eggs) of *Trichuris muris* (Edinburgh strain) suspended in 200 μl water ($n = 7$), or with 200 μl water alone as a control ($n = 5$). Animals were housed at 22 °C ± 1 °C with 65% humidity and a 12 h light-dark cycle with free access to food and water. Faecal pellets were collected directly into 750 μl pre-chilled RNA*later*® (Ambion) and stored at 4 °C prior to the extraction of total RNA (within 24 h) using the mirVana™ miRNA Isolation Kit (Invitrogen). Extractions were repeated from the same collection of faecal pellets within 24 h if the initial extraction did not yield clear and distinct ribosomal bands measured by the

**Fig. 3 | miRNAs differentially expressed by *T. muris* infection regulate fibrosis.**
**a** Ingenuity pathway analysis was used to identify the top 16 enriched pathways of the mRNA targets of the faecal miRNAs differentially expressed on day 35 of chronic *T. muris* infection. Targets represent the number of mRNA targets of the differentially expressed miRNAs in each pathway. The mRNA targets were pre-filtered to only include those with a confidence level of 'High predicted' or 'Experimentally observed' prior to pathway analysis. *P* value is a result of Fisher's exact test (right-tailed). **b**–**d** 3T3 fibroblasts were cultured in TGF-β in the presence of 50 nM of negative control miRNA (miR-ctrl), miR-29a mimic or inhibitor, miR-200c mimic or inhibitor, or miR-26b mimic or inhibitor for 36 h. **b** RNA was harvested and the expression of *Col1* quantified by qPCR relative to the expression in the untreated untransfected cells in each biological replicate. Expression levels were normalised to *Gapdh* using the ΔΔCT method. Statistical significance was calculated by one-way ANOVA with Geisser-greenhouse correction and Dunnet's multiple comparisons test. The mean and SD are displayed. $n = 3$ or $n = 4$ biological replicates for each condition. **c** Transfected fibroblasts were fixed after 36 h, stained for collagen I (yellow) and a nuclear stain (blue) and imaged by confocal microscopy using a 63x oil immersion objective. Scale bars are 50 μm. **d** Luminex cytokine profiling of fibroblast cell culture supernatant after 36 h of TGF-β treatment and transfections. Statistical analysis is a result of a one-way ANOVA with Holm–Šídák's multiple comparisons test. The mean and SD are displayed. $n = 3$ biological replicates for each condition.

TapeStation electropherogram. Total RNA was stored at −80 °C prior to preparation for sequencing. Small RNA sequencing was performed on 31 samples, from the infected and naïve groups at day 21 and day 35 and additionally from the infected group prior to infection on day 0.

## Library building and small RNA sequencing
Small RNA sequencing libraries were built using the NEBnext small RNA library prep set for Illumina (New England Biolabs) following the manufacturer's instructions. 500 ng total RNA per sample was used as input with undiluted SR adaptors and RT primers. Amplification was performed with 12 PCR cycles. Size selection was performed using the Novex 6% TBE gel (Invitrogen, Life Technologies). Samples were pooled and 76 bp paired-end sequencing was performed on the Illumina Hiseq4000 at the University of Manchester Genomic Technologies Core Facility. The faecal small RNA sequencing data generated during the course of this study have been deposited in the NCBI Gene Expression Omnibus database, under accession number GSE267555 ($n = 31$).

## Collection of data for meta-analysis
The inclusion criteria for studies in the meta-analysis was, to the best of our knowledge, all studies that have published publicly available faecal small RNA sequencing data from C57BL/6 mice during the study period (up until October 2024). Fastq files for published data were obtained through the following NCBI bioproject accession numbers: Tomkovich et al., PRJNA422621 ($n = 21$)[14], He et al., PRJNA695265 ($n = 11$)[15], and Liu et al., PRJNA563061 ($n = 15$)[16].

## Analysis of small RNA sequencing data
The forward read of each read pair for paired-end data was used for analysis of all collected and generated datasets. Cutadapt version 1.8[79] was used to remove sequencing adaptors, filter for quality, and for reads of 18–26 nucleotides in length. Small RNA reads were firstly mapped to tRNA and rRNA with 0 mismatches using bowtie1 version 1.1.1[80], and all hits were excluded from further analysis. Remaining reads were mapped to the mouse genome (mm10) using Bowtie1 and the following parameters: -v 1 -m 10 -S -a --best --strata. Mouse miRNAs were quantified using the featureCounts function of Subread version 1.6.0[81] using the mmu.gff3 annotation file from miRbase[82]. For miRNAs annotated in multiple locations of the mouse genome, the locus with the highest number of small RNA reads mapped to it was used for further analysis. Reads were normalised using the counts() function of DESeq2 which adjusts for differences in sequencing depth and RNA composition by calculating size factors. The bioinformatics pipeline developed herein is publicly available and can be found at https://github.com/elayton13/faecalmiRseq.

## Differential expression and pathway analysis
Differential expression analysis was performed using the DESeq function of DESeq2[83]. The Wald test was used to calculate statistical significance and Benjamini-Hochberg false discovery corrected

miRNAs with an adjusted *p* value < 0.05 were considered differentially expressed. Functional analysis was performed on the predicted mRNA targets of the differentially expressed miRNAs at day 35 post-infection using Ingenuity Pathway Analysis (Qiagen). The mRNA targets were pre-filtered to only include those with a confidence level of 'High predicted' or 'Experimentally observed' prior to pathway analysis. Fisher's exact test (right-sided) was used to identify over-represented pathways in the mRNA targets.

## Cell culture and cytokine profiling
NIH-3T3 mouse fibroblasts (ATCC CRL-1658) were seeded overnight prior to treatment with 10 ng/mL recombinant mouse Transforming Growth Factor β1 (TGF-β, R&D systems, catalogue ID: 7666-MB-005/CF) and simultaneously transfected with 50 nM of mirVana™ mimics, or inhibitors (Life Technologies), or a cy3-labelled negative control miRNA (Dharmacon, catalogue ID: CN-001000-01-05). Cell culture supernatants were removed at the end of the experiment and stored at −80 °C prior to cytokine analysis using the MILLIPLEX MAP Mouse Cytokine/Chemokine Magnetic Bead Panel (catalogue ID: MYCTO-MAG-70K).

## Confocal imaging
NIH-3T3 cells were incubated for 36 h in 10 ng/mL TGF-β and transfected as described (see above). Cells were fixed in 4% paraformaldehyde (PFA) and permeabilised with PBST prior to staining with primary antibodies diluted in PBST at 1:2000 for collagen I (Abcam, catalogue ID: ab270993) and nuclear staining with Hoechst. NIH-3T3 cells were imaged on the LSM800 confocal microscope (Carl Zeiss) using a 63× oil immersion objective.

## RNA isolation and qPCR
Total RNA was isolated from fibroblasts at the end of the 36-h transfection experiment using the RNeasy Micro Kit (Qiagen) and stored at −80 °C. cDNA was synthesised using the SuperScript™ III First-Strand Synthesis System (Invitrogen) and GoTaq® qPCR Master Mix (Promega) used for quantification on the QuantStudio 6 Flex System (Applied Biosystems). Relative quantification was performed using the ΔΔCT method normalised to mouse *Gapdh*.

## Hyperion imaging mass cytometry
Sections of caecum tissue were initially deparaffinized using a standard xylene and ethanol gradient to rehydrate the samples. Microwave antigen retrieval was performed in Tris EDTA buffer (pH 9.0) for 20 min. Non-specific proteins were blocked in 3% (w/v) bovine serum albumin (BSA) in 5% (v/v) goat serum with 0.05% (v/v) Tween for 1 h at room temperature followed by blocking endogenous avidin/biotin (Biolegend Blocking System). Metal-conjugated primary antibodies, diluted 1:50 for Collagen VI (Abcam, catalogue ID: ab229450) and Heparan Sulphate (AmsBio, catalogue ID: 270255-1), 1:200 for Collagen I (Standard Biotools, catalogue ID: 3169023D), 1:150 for alpha-SMA (Standard Biotools, catalogue ID 3141017D) and 1:100 for Hyaluronic acid binding protein-biotinylated (Merck,

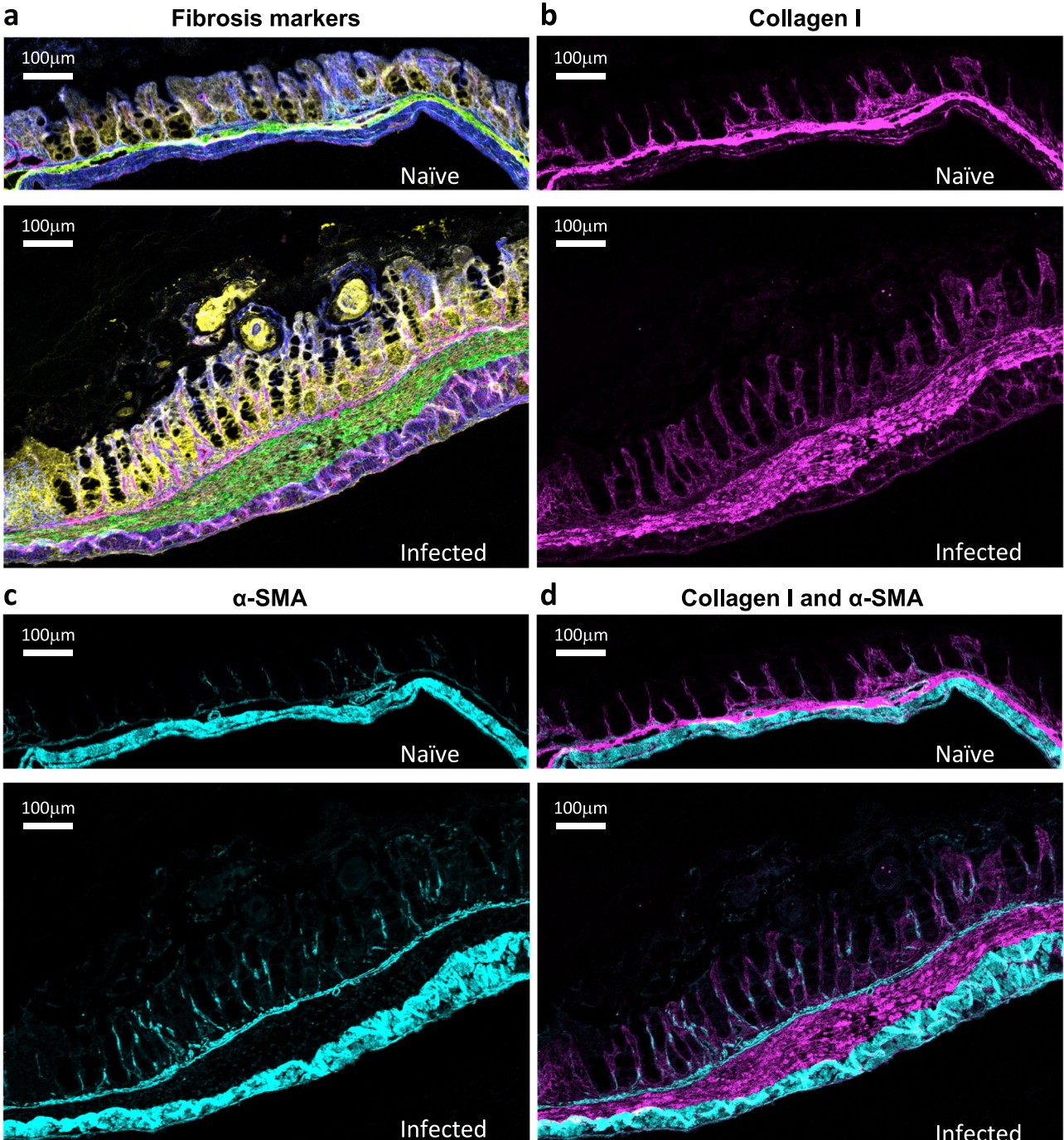

**Fig. 4 | Evidence of fibrosis in the caecum of *T. muris*-infected mice.** Ceca from a naïve and a chronically infected C57BL/6 mouse on day 42 post-gavage of water or low-dose *T. muris* eggs were randomly selected for assessment with Hyperion® imaging mass cytometry. **a** Ceca stained for Collagen VI (blue), Collagen I (green), Hyaluronan (yellow) and Heparan sulphate (magenta). **b** For Collagen I (magenta). **c** For α-SMA (cyan). **d** For Collagen I (magenta) and α-SMA (cyan).

catalogue ID: 385911) were prepared in PBS supplemented with 0.5% (w/v) BSA, and tissue sections stained overnight at 4 °C. The tissue sections were washed with PBS and incubated with a secondary metal-conjugated anti-biotin antibody (Standard Biotools, catalogue ID: 3170003B) at a dilution of 1:150 for 1 h at room temperature, followed by washing in PBS (further details can be found in Supplementary Table 1). Sections were stained with DNA intercalator (1:800, Standard Biotools, catalogue ID: 201192B) for 20 min. Imaging was performed on an imaging mass cytometer (Hyperion, Standard Biotools) and carried out in the Flow Cytometry core facility at the

University of Manchester. The Hyperion mass cytometer was funded by the BBSRC (BB/S019324/1).

## Statistics and reproducibility

Statistical analysis was carried out in GraphPad Prism (version 10.0.1 for MacOS X, GraphPad Software, San Diego, California USA). Sample sizes for faecal miRNA sequencing were based on previously published literature. No data were excluded from the analyses. Allocation of mice to experimental groups was randomised. Investigators were not blinded to allocation during experiments and outcome assessment.

**Reporting summary**

Further information on research design is available in the Nature Portfolio Reporting Summary linked to this article.

## Data availability

Sequencing data generated during the course of this study is deposited under the Gene Expression Omnibus accession number: GSE267555. Fastq files for previously published data used in the meta-analysis were obtained through the following NCBI bioproject accession numbers: Tomkovich et al., PRJNA422621 ($n=21$)[14], He et al., PRJNA695265 ($n=11$)[15], and Liu et al., PRJNA563061 ($n=15$)[16]. Source data are provided with this paper.

## Code availability

The bioinformatics pipeline developed is publicly available and can be found at https://github.com/elayton13/faecalmiRseq.

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

## Acknowledgements
R.K.G., S.T. and A.J.B. were supported by Wellcome Trust Investigator Award Z10661/Z/18/Z and the Wellcome Centre for Cell Matrix Research Grant 088785/Z/09/Z. The Hyperion mass cytometer was funded by the BBSRC grant BB/S019324/1. E.L. was supported by the joint A*STAR-University of Manchester PhD programme.

## Author contributions
Conceptualisation: E.L., S.G.J., R.K.G., A.M.F., I.S.R.; methodology: E.L., S.G., E.Y., W.Y.O., T.S., S.T., V.B.A., S.G.J., R.K.G., A.M.F., I.S.R.; investigation: E.L., W.Y.O., T.S., A.J.B., S.T., V.B.A. Writing—original draft: E.L., R.K.G., A.M.F., I.S.R.; writing—reviewing and editing: E.L., S.G., E.Y., W.Y.O., T.S., A.J.B., S.T., V.B.A., S.G.J., R.K.G., A.M.F., I.S.R.; visualisation: E.L. and E.Y.; funding acquisition: R.K.G., A.M.F., I.S.R.; resources: R.K.G., A.M.F., I.S.R.; supervision: S.G.J., R.K.G., A.M.F., I.S.R.

## Competing interests
The authors declare no competing interests.
