## [Transparent Peer Review file · Nature Communications]

An Optimised Faecal microRNA Sequencing Pipeline Reveals Fibrosis in *Trichuris muris* Infection

Corresponding Author: Professor Ian Roberts

Version 0:

Reviewer comments:

Reviewer #1

(Remarks to the Author)

The manuscript "An Optimised Faecal microRNA Sequencing Pipeline Reveals Fibrosis in *Trichuris muris* Infection" is an interesting work on the detection of miRNAs in mouse feces in general and their changes in response to *T. muris* infection. The methodology for isolating miRNAs from feces is novel and has the potential for development for diagnosis and prognosis of various pathologies. However, I find it noteworthy that fibroblasts are used in "in vitro" studies rather than epithelial cells, as the latter are the ones in contact with the parasite. Additionally, the focus is on the role of miR-29a and miR-200c in collagenogenesis, and there is no analysis or discussion of their role in cell proliferation and differentiation, among others. Both proliferation and differentiation are important processes in the response to pathogens. In Figure 4, changes in glycosaminoglycans and proteoglycans are evident; however, these changes are not discussed in detail. A problem in interpreting the miRNA data is the fact that one miRNA regulates many mRNAs, and an mRNA is in turn regulated by several miRNAs. Therefore, associating only miR-29a and miR-200c (results obtained in vitro) with the fibrosis observed in response to *T. muris* (in vivo) seems a bit "adventurous" to me.

(Remarks on code availability)

I could easily access and run the application

Reviewer #2

(Remarks to the Author)

The authors perform sRNA sequencing of mouse faecal samples and compare samples taken from mice infected with *T. muris* at two time-points to equivalent uninfected samples as well as a baseline. They identify miRNAs that change during infection, leading them to propose a hypothesis related to fibrosis and wound healing. This is tested in vitro in a fibroblast cell line using miRNA mimics and inhibitors as well as in vivo comparing caecal tissue from infected and control mice by immunostaining. Overall, the work is well performed and provides a proof-of-principle of how faecal miRNA analysis can be used to initiate functional hypotheses, and suggests further applications related to non-invasive diagnostics.

I have a series of questions and suggestions that I list below, followed by some minor comments:

- Results refer to "mean normalised counts" and "Normalised miRNA reads", yet Methods do not describe how these were calculated (are they mean CPM? Mean log2CPM? Something else?).
- Provide PCA/MDS or equivalent of all samples, so the reader can better understand how individual samples are related.
- Provide, as supplementary, a table with the counts of all the miRNAs across all the samples, that were used for expression analyses. Also provide results of differential expression analyses (fold-changes, adj p-values, etc).
- Provide, as supplementary, a table of the miRNA's mRNA targets that were used for pathway analysis.
- Lines 270-272: Could part of the trends observed for family members of let-7 and miR-200 be due to the same read (sequence) being shared between them? Since featureCounts considers reads of different lengths and any overlap, this is possible. Can you check how many reads are shared between family members? Or conversely, how if uniquely-mapping reads show the same trend?
- Why not use 1 mismatch when mapping against rRNA and tRNA? A potential false-positive was manually excluded (Line 332). Perhaps more reads like this remain.
- Why bowtie 1.1.1? Is this a typo? It was released ten years ago, and still had many bugs that have since been corrected.

Current version is 1.3.1, and was released in 2021.

- Why were only male mice used for these experiments? Is there more variability between faecal samples from different sexes than between treatments? Can this be commented on?

Minor comments:

- Figure 2b key colour is not adequately described. It probably is not ranking (as the key title suggests), since the space between ranking values is not constant. Perhaps the colour is determined by expression values, and different ranking positions are overlaid?
- Figure 3b. Please show y-axis in log₂ scale, so we can equally see reductions and increases of expression.
- Figure 4 is a representative image for caecum tissue staining. Where other sections taken? Can more images be provided as supplementary?
- Line 120: the double dash for --strata has been autocorrected.
- Line 510: "effective exclusion of rRNA fragments prior during library preparation". Change "during" -> "to"
- Line 547: references 61 and 62 highlight that a certain Argonaute protein is present within EVs, but this is not a miRNA-binding Argonaute. In addition, reference 62 highlights that miRNAs are a very minor component of EVs. So, I'm not sure how these references help the current discussion.

Pipeline comments:

- For the removal of tRNA and rRNA steps, I would recommend using the --un <file> argument of bowtie1 directly, to output unaligned reads. Currently two additional invocations of samtools are required, which is not efficient.
- For the final alignment to the mouse genome, it is more efficient to pipe the SAM output of bowtie directly to samtools for sorting and conversion to BAM.
- Could you provide in the same github the R script that was used for expression analysis?

(Remarks on code availability)

The code is a simple bash script with a few command lines. These commands are quite straightforward and the code simply helps to clarify the exact command line arguments that the authors have used for their analysis (up until miRNA count quantification). The code does not include the steps of count normalisation or differential expression analysis. I have some minor suggestions/questions about it, included in my comments to the authors.

Reviewer #3

(Remarks to the Author)

Dear Dr Layton and Colleagues,

I have had the pleasure of reviewing your manuscript entitled "An Optimised Faecal microRNA Sequencing Pipeline Reveals Fibrosis in *Trichuris muris* Infection". This study addresses important questions regarding optimal methods of faecal miRNA extraction and sequencing, and the reproducibility of the sequencing results.

This study presents the following key findings:

1. An approach for extracting miRNA sequences from murine faecal samples with comparison to approaches used in other studies (Table 1)
2. Demonstrating that a subset of murine faecal miRNA profiles are relatively conserved across studies (Figure 1) and over time when following healthy wild-type mice for up to 35 days (Figure 2)
3. Chronic *T. muris* infection induces changes in host faecal miRNA expression which promote intestinal fibrosis and may regulate the inflammatory/anti-inflammatory immune response (Figure 3)

Together points 1 and 2 above demonstrate that sequencing of faecal miRNAs can produce consistent findings in the face of common sources of variation such as differing laboratory practices and extended time courses often required during challenge studies. This is particularly important as miRNAs are notoriously difficult to sequence due to their small size and potential for degradation. Point 3 demonstrates the utility of the authors miRNA sequencing pipeline for answering questions of biological and clinical significance. In particular they show that differential miRNA expression coupled with pathway analysis can be used to generate testable hypotheses toward understanding drivers of disease pathology. In this case, the role of miRNA-200c and miR-29a in *T. muris*-induced intestinal fibrosis, which is a marker of poor clinical outcome.

The authors validate these findings the Imaging Mass Cytometry staining of *T. muris* infected Caecal tissues demonstrating patterns of fibrosis consistent with the prior miRNA analyses.

Given the paucity of studies examining faecal miRNA extraction methods and consistency of miRNA profiles, as well as the applications revealing a potentially novel role of miRNAs in *T. muris* infection, which is of high clinical relevance, I am happy for the study to be published, provided the revisions below can be addressed.

Revisions:

Please justify limiting the inclusion of studies to January 2023. I understand if this was the prior study period, however as of

writing it is August 2024 and there is likely newer studies available which could be included.

Line 188 - Please provide reference for "Due to the high proportion of small RNAs in faecal samples relative to other tissues..."

Line 163 - Methods Imaging Mass Cytometry section. Please also provide antibody concentrations used for staining. In particular the concentration used after metal conjugation. Also provide the panel itself that was used (i.e the metal isotopes to which each antibody was tagged)

Table 1 - Please provide description of calculation for Normalised miRNA reads and any other metrics not described. Also for the sake of presentation you may wish to change this to a graphical format and move the table to the supplementary. It is not at all obvious looking at the table, what the reader should take away from it. Only the key results should be in the main text.

Could you please provide graphical data with statistical comparisons for all measurements referenced in text between lines 327 and 337.

Figure 4 - Please clearer images and zoomed areas to demonstrate the results described for Figure 4a. Also as single images presumably from one sample each (infected vs Naive) are provided please provide either additional donor data in the supplementary or quantification in the primary figures.

Please contextualise the results described between lines 434 and 441. A description of the staining was provided but not the relevance of the markers for the result itself.

Line 490 - The opening line of the discussion is a bit of non-sequitur. The main subject of the paper is not the development of a screening tool for clinical use. This might be more suited for the end of the discussion.

(Remarks on code availability)

Version 1:

Reviewer comments:

Reviewer #1

(Remarks to the Author)

The authors included the suggestions of the reviewer in the new version of the manuscript. Therefore I recommend accepting the manuscript in the present form

(Remarks on code availability)

I have revised the code and had no problem running it. I believe that the code is usable for the community

Reviewer #2

(Remarks to the Author)

I have read the Response to Reviewers, as well as the changes made to the manuscript. I feel that the authors have adequately addressed the comments from the reviewers. I have no further suggestions to add.

(Remarks on code availability)

The bash code script remains the same, but importantly the R script for performing the DESeq2 analysis of the miRNA counts has been added.

Reviewer #3

(Remarks to the Author)

Thank you for providing the revised manuscript. I have reviewed the document and the rebuttal comments and believe the suggested revisions have been adequately addressed by the authors. The only exception is Figure 4, where I was unable to find the donor numbers within the figure legend as stated in the rebuttal comments.

(Remarks on code availability)

Response to Reviewers

We would like to thank the reviewers for the constructive reviews which have resulted in a much-improved manuscript. We have addressed all of the points raised (see below) and now feel that the manuscript is suitable for publication. Please see the PDF version of our revised manuscript for the line numbers referred to herein. A Microsoft word version with tracked changes is also supplied in our revised submission.

REVIEWER #1

Reviewer Comment:

"The manuscript 'An Optimised Faecal microRNA Sequencing Pipeline Reveals Fibrosis in *Trichuris muris* Infection' is an interesting work on the detection of miRNAs in mouse feces in general and their changes in response to *T. muris* infection. The methodology for isolating miRNAs from feces is novel and has the potential for development for diagnosis and prognosis of various pathologies. However, I find it noteworthy that fibroblasts are used in "in vitro" studies rather than epithelial cells, as the latter are the ones in contact with the parasite."

We thank the reviewer for their supportive comments and suggestions to improve the manuscript. Our decision to use fibroblasts was based on the key role that these cells play in the fibrotic response. Fibrosis is characterised by the excessive deposition of extracellular matrix components, predominantly produced by fibroblasts. While epithelial cells are indeed the first to encounter the parasite, fibroblasts are central to the downstream fibrotic response, which is the primary focus of our study. Furthermore, available intestinal epithelial cell lines, such as MODE-K, are derived from the small intestine, which is not representative of the caecal environment. We agree that future studies could benefit from exploring the role of miRNAs in epithelial cells during *T. muris* infection, particularly in the early stages of infection. We believe this is beyond the scope of the current study which focuses on fibroblast-driven fibrosis but have added this point to the discussion. Please refer to lines 636-638.

Reviewer Comment:

"Additionally, the focus is on the role of miR-29a and miR-200c in collagenogenesis, and there is no analysis or discussion of their role in cell proliferation and differentiation, among others. Both proliferation and differentiation are important processes in the response to pathogens."

We agree and recognize that miR-29a and miR-200c are involved in various cellular processes, including cell proliferation and differentiation, the integrity of the epithelial barrier and in the polarisation of T cell and macrophage responses - each of which are important processes in the pathogen response. Our primary focus was on collagenogenesis due to its direct relevance to fibrosis, the main pathological feature indicated by our pathway analysis of the differentially expressed faecal miRNAs in *T. muris* infection. We have expanded our discussion to include further detail on other roles of these miRNAs that may be relevant to *T. muris* infection. Please see lines 620-626.

Reviewer Comment:

"In Figure 4, changes in glycosaminoglycans and proteoglycans are evident; however, these changes are not discussed in detail."

We have added additional context to the description of these markers and their relevance to fibrosis in the results section. Please see lines 468-477.

Reviewer Comment:

"A problem in interpreting the miRNA data is the fact that one miRNA regulates many mRNAs, and an mRNA is in turn regulated by several miRNAs. Therefore, associating only miR-29a and miR-200c (results obtained in vitro) with the fibrosis observed in response to *T.*

muris (in vivo) seems a bit 'adventurous' to me."

The primary goal of our study was to illustrate the utility of our optimised faecal miRNA sequencing pipeline as a method for developing testable hypotheses. Our focus on miR-29a and miR-200c was driven not only by their documented roles in fibrosis, but also by the fact that they were among the most differentially expressed miRNAs in our dataset. While our findings suggest a potential link between these miRNAs and fibrosis in *T. muris* infection, we agree that further studies are needed to validate their *in vivo* relevance. We have expanded the discussion on fibrosis development in helminth infection and provided additional evidence supporting a potential role for miR-29a in helminth-induced fibrosis. In the revised manuscript, we also highlight the need for further research to fully elucidate the role of these miRNAs in *T. muris*-induced fibrosis and pathology. Please see discussion lines 628–638.

REVIEWER #2

The authors perform sRNA sequencing of mouse faecal samples and compare samples taken from mice infected with *T. muris* at two time-points to equivalent uninfected samples as well as a baseline. They identify miRNAs that change during infection, leading them to propose a hypothesis related to fibrosis and wound healing. This is tested in vitro in a fibroblast cell line using miRNA mimics and inhibitors as well as in vivo comparing caecal tissue from infected and control mice by immunostaining. Overall, the work is well performed and provides a proof-of-principle of how faecal miRNA analysis can be used to initiate functional hypotheses, and suggests further applications related to non-invasive diagnostics.

We thank the reviewer for their supportive comments for the manuscript and have addressed the questions and comments below.

I have a series of questions and suggestions that I list below, followed by some minor comments:

- Results refer to “mean normalised counts” and “Normalised miRNA reads”, yet Methods do not describe how these were calculated (are they mean CPM? Mean log₂CPM? Something else?).

We have updated the methods to include that DESeq2 was used for normalisation. DESeq2 uses size factor normalisation to correct for differences in sequencing depth. The mean normalised counts are for the average normalised counts for each miRNA across the animals in each group. Please see lines 128-130.

- Provide PCA/MDS or equivalent of all samples, so the reader can better understand how individual samples are related.

We have added a PCA plot to the supplementary materials and directed the reader to this with lines 362-365 (Supplemental Figure 2).

- Provide, as supplementary, a table with the counts of all the miRNAs across all the samples, that were used for expression analyses. Also provide results of differential expression analyses (fold-changes, adj p-values, etc).

This has been added as an additional supplementary file, Layton_supplementary_table_2.csv

- Provide, as supplementary, a table of the miRNA's mRNA targets that were used for pathway analysis.

This has been added as an additional supplementary file, Layton_supplementary_table_3.csv

- Lines 270-272: Could part of the trends observed for family members of let-7 and miR-200 be due to the same read (sequence) being shared between them? Since featureCounts considers reads of different lengths and any overlap, this is possible. Can you check how many reads are shared between family members? Or conversely, how if uniquely-mapping reads show the same trend?

We have performed additional analysis of our data to ensure that the trend that we see for each miRNA family over time in the naïve mice is not due to the sharing of reads. We have added a figure to the supplemental information (supplemental figure 3) that reproduces the original analysis, whilst only counting the uniquely mapped reads for the let-7 and miR-200 miRNAs. This confirmed that the trend remains the same, and the reader has been directed to this additional analysis by lines 286-289.

- Why not use 1 mismatch when mapping against rRNA and tRNA? A potential false-positive was manually excluded (Line 332). Perhaps more reads like this remain.

miR-6239 is currently annotated as a miRNA in miRbase and as such others have claimed that it is a true miRNA. However, we have taken the conservative approach to exclude it from our analysis. It is possible that more reads remain that could map to RNAs other than miRNAs either exactly or with 1 mismatch. However, we have carefully inspected all mappings to the miRNAs that we discuss in the manuscript, and we are confident in their status. Indeed, after the exclusion of miR-6239 we have focussed on the study of very well validated high confidence miRNAs. We therefore have no concern that other miRNAs mentioned in this study are potential false positives.

- Why bowtie 1.1.1? Is this a typo? It was released ten years ago, and still had many bugs that have since been corrected. Current version is 1.3.1, and was released in 2021. Bowtie 1.1.1 has been widely used as an aligner for the analysis of miRNA sequencing data and remains a robust and reliable choice for this purpose. Whilst 1.3.1 included minor bug fixes and performance improvements over previous versions, it did not introduce new features. However, we have added a note to the github repo to highlight that a newer version is available if problems are experienced with bowtie 1.1.1.

- Why were only male mice used for these experiments? Is there more variability between faecal samples from different sexes than between treatments? Can this be commented on? We thank the reviewer for raising this important question and are happy to clarify our rationale further. The decision to use male mice was made to minimise biological variability, ensuring a more controlled assessment of the roles of miRNAs in *T. muris* infection. While sex differences can indeed influence infection outcomes, the fundamental mechanisms governing susceptibility and resistance remain consistent across sexes. A comprehensive investigation into the impact of sex hormones and chromosomal differences on faecal miRNAs, though highly valuable, would represent a substantial and distinct project beyond the scope of the current study. This hasn't been commented on to maintain the flow and conciseness of the current discussion.

Minor comments:

- Figure 2b key colour is not adequately described. It probably is not ranking (as the key title suggests), since the space between ranking values is not constant. Perhaps the colour is determined by expression values, and different ranking positions are overlaid?

We have modified figure 2b to improve the clarity of the legend.

- Figure 3b. Please show y-axis in log₂ scale, so we can equally see reductions and increases of expression.

We have replotted the data with a log₂ scale on the y-axis (see below) and upon comparison feel that the original plot remains a better representation of the data.

- Figure 4 is a representative image for caecum tissue staining. Where other sections taken? Can more images be provided as supplementary?

Sections were only taken from the caecum as this is where *T. muris* is located and where pathology occurs.

- Line 120: the double dash for --strata has been autocorrected.

Thank you, we have amended this.

- Line 510: “effective exclusion of rRNA fragments prior during library preparation”. Change “during” -> “to”

Thank you, we have amended this.

- Line 547: references 61 and 62 highlight that a certain Argonaute protein is present within EVs, but this is not a miRNA-binding Argonaute. In addition, reference 62 highlights that miRNAs are a very minor component of EVs. So, I’m not sure how these references help the current discussion.

Reference 62 of the original manuscript was cited to support the presence of argonaute in helminth EVs. Whilst this paper does also describe miRNAs to be a minor component of EVs, this was only demonstrated in EVs from *H. bakeri*. Later research from Buck and colleagues (2020) was cited in the following sentence (reference 63 of the original manuscript), and specifically compared the miRNA content of *H. bakeri* and *T. muris* EVs. This work found that in *T. muris* EVs, miRNAs were more abundant than in *H. bakeri* EVs. We have edited the first sentence to also include this reference.

Whilst the argonaute referred to in the cited references may not bind to miRNAs, other studies have identified nematode argonaute proteins such as ALG-1 and ALG-2 that do bind miRNAs and mediate their function. Furthermore, nematode miRNAs found in EVs have been shown to be able to repress the transcription of host mRNAs to modulate host immunity in a paper by Buck and colleagues (2014) titled “Exosomes secreted by nematode parasites transfer small RNAs to mammalian cells and modulate innate immunity”. The role of the worm-specific argonaute in this process remains unclear. We have discussed this at lines 647-648 and clarified that further studies are required to determine the molecular basis of these miRNAs in such inter-species interactions.

Pipeline comments:

- For the removal of tRNA and rRNA steps, I would recommend using the `--un <file>` argument of bowtie1 directly, to output unaligned reads. Currently two additional invocations of samtools are required, which is not efficient. For the final alignment to the mouse genome,

it is more efficient to pipe the SAM output of bowtie directly to samtools for sorting and conversion to BAM.

The pipeline is a simple bash script that is designed to be accessible to a wide audience, particularly those with limited bioinformatics experience. The reviewer's suggestion could indeed be more efficient, but it does not affect the results in any way, and the current script enables users to easily interrogate the intermediate rRNA and tRNA files generated if desired.

- Could you provide in the same github the R script that was used for expression analysis? Thank you for the suggestion, we have now also added the R script used for differential analysis to the same github.

Reviewer #2 (Remarks on code availability):

The code is a simple bash script with a few command lines. These commands are quite straightforward and the code simply helps to clarify the exact command line arguments that the authors have used for their analysis (up until miRNA count quantification). The code does not include the steps of count normalisation or differential expression analysis. I have some minor suggestions/questions about it, included in my comments to the authors.

Reviewer #3

Dear Dr Layton and Colleagues,

I have had the pleasure of reviewing your manuscript entitled “An Optimised Faecal microRNA Sequencing Pipeline Reveals Fibrosis in *Trichuris muris* Infection”. This study addresses important questions regarding optimal methods of faecal miRNA extraction and sequencing, and the reproducibility of the sequencing results.

This study presents the following key findings:

1. An approach for extracting miRNA sequences from murine faecal samples with comparison to approaches used in other studies (Table 1)
2. Demonstrating that a subset of murine faecal miRNA profiles are relatively conserved across studies (Figure 1) and over time when following healthy wild-type mice for up to 35 days (Figure 2)
3. Chronic *T. muris* infection induces changes in host faecal miRNA expression which promote intestinal fibrosis and may regulate the inflammatory/anti-inflammatory immune response (Figure 3)

Together points 1 and 2 above demonstrate that sequencing of faecal miRNAs can produce consistent findings in the face of common sources of variation such as differing laboratory practices and extended time courses often required during challenge studies. This is particularly important as miRNAs are notoriously difficult to sequence due to their small size and potential for degradation. Point 3 demonstrates the utility of the authors miRNA sequencing pipeline for answering questions of biological and clinical significance. In particular they show that differential miRNA expression coupled with pathway analysis can be used to generate testable hypotheses toward understanding drivers of disease pathology. In this case, the role of miRNA-200c and miR-29a in *T. muris*-induced intestinal fibrosis, which is a marker of poor clinical outcome.

The authors validate these findings the Imaging Mass Cytometry staining of *T. muris* infected Caecal tissues demonstrating patterns of fibrosis consistent with the prior miRNA analyses.

Given the paucity of studies examining faecal miRNA extraction methods and consistency of miRNA profiles, as well as the applications revealing a potentially novel role of miRNAs in *T. muris* infection, which is of high clinical relevance, I am happy for the study to be published, provided the revisions below can be addressed.

Thank you for your detailed overview of our study and the supportive comments. We have addressed your suggested revisions below.

Revisions:

Please justify limiting the inclusion of studies to January 2023. I understand if this was the prior study period, however as of writing it is August 2024 and there is likely newer studies available which could be included.

We acknowledge the reviewer's concern regarding the inclusion period for the studies in our meta-analysis. Initially, we limited the inclusion of studies to January 2023, as this was the period during which we conducted this part of our research. We have now performed an additional literature search, detailed below.

On October 26th 2024, we conducted a PubMed search using the terms "mice AND faecal OR fecal AND miRNA," which yielded 348 results. Despite this, no further studies analysing murine faecal miRNA in C57BL/6 mice using small RNA sequencing were identified.

We did identify more recent studies employing murine faecal miRNA sequencing, such as Yan *et al.* (2024) titled "*Chlorogenic acid ameliorates intestinal inflammation via miRNA-microbe axis in db/db mice,*" and Li *et al.* (2023) titled "*Host miR-129-5p reverses effects of ginsenoside Rg1 on morphine reward, possibly mediated by changes in B. vulgatus and serotonin metabolism in the hippocampus.*"

Yan *et al.*, used db/db mice and performed miRNA sequencing of colonic contents and so this is not directly comparable to our data on faecal miRNAs from C57BL/6 mice. For Li *et al.*, they provided accession numbers their raw faecal miRNA sequencing data in BALB/c mice, unfortunately we were unable to find the corresponding uploads for this data on NCBI or SRA.

We have updated our inclusion criteria to reflect that, to the best of our knowledge, all studies to date that have performed small RNA sequencing of faecal samples in C57BL/6 mice have been included in our analysis.

Line 188 - Please provide reference for "Due to the high proportion of small RNAs in faecal samples relative to other tissues..."

This comment was based on data obtained during the optimisation of the library building protocol. A tapestation image illustrating these differences has been added as an additional panel to Supplemental figure 1.

Line 163 - Methods Imaging Mass Cytometry section. Please also provide antibody concentrations used for staining. In particular the concentration used after metal conjugation. Also provide the panel itself that was used (i.e the metal isotopes to which each antibody was tagged)

We have now included this information in the revised submission (Supplemental Table 1).

Table 1 - Please provide description of calculation for Normalised miRNA reads and any other metrics not described. Also for the sake of presentation you may wish to change this to a graphical format and move the table to the supplementary. It is not at all obvious looking at the table, what the reader should take away from it. Only the key results should be in the main text.

We have clarified the normalisation methods as requested by reviewer 1. However, we feel that the table remains the best representation of the qualitative and quantitative data that we present to describe the sequencing protocols and outputs.

Could you please provide graphical data with statistical comparisons for all measurements referenced in text between lines 327 and 337.

We have added a supplementary file containing this information to our manuscript submission, as requested by reviewer 1.

Figure 4 - Please clearer images and zoomed areas to demonstrate the results described for Figure 4a. Also as single images presumably from one sample each (infected vs Naive) are provided please provide either additional donor data in the supplementary or quantification in the primary figures.

The images provided are of the maximum resolution for Hyperion imaging, however we have also added zoomed areas to the supplemental information to improve clarity (Supplemental Figure 6). Further detail of the fibrotic markers has also been added to the results section, and the legend has been updated to clarify the donor data.

Please contextualise the results described between lines 434 and 441. A description of the staining was provided but not the relevance of the markers for the result itself.

We agree that additional context for the Hyperion markers used will aid in the understanding of our methodology. We have incorporated this to the revised manuscript, please see lines 468-477 of the results section.

Line 490 - The opening line of the discussion is a bit of non-sequitur. The main subject of the paper is not the development of a screening tool for clinical use. This might be more suited for the end of the discussion.

We have edited the first few lines of the discussion to address this (Lines 550-553).

Reviewer #3 (Remarks to the Author):

Thank you for providing the revised manuscript. I have reviewed the document and the rebuttal comments and believe the suggested revisions have been adequately addressed by the authors. The only exception is Figure 4, where I was unable to find the donor numbers within the figure legend as stated in the rebuttal comments.

In the final version of the manuscript, we amended the legend of Figure 4 to address this point making it clear that that a single animal was randomly selected from both the naïve and infected groups for Hyperion analysis. Please find the modified Figure legend below.

Figure 4. Evidence of Fibrosis in the Caecum of *T. muris* Infected Mice. Ceca from a naïve and a chronically infected C57BL/6 mouse on Day 42 post-gavage of water or low-dose *T. muris* eggs were randomly selected for assessment with Hyperion® imaging mass cytometry. **a** Ceca stained for Collagen VI (blue), Collagen I (green), Hyaluronan (yellow) and Heparan sulphate (magenta). **b** For Collagen I (magenta). **c** For α -SMA (cyan). **d** For Collagen I (magenta) and α -SMA (cyan).